# DETECTION-ORIENTED IMAGE-TEXT PRETRAINING FOR OPEN-VOCABULARY DETECTION

## ABSTRACT

We present a new open-vocabulary detection approach based on detection-oriented image-text pretraining to bridge the gap between image-level pretraining and open-vocabulary object detection. At the pretraining phase, we replace the commonly used classification architecture with the detector architecture, which better serves the region-level recognition needs of detection by enabling the detector heads to learn from noisy image-text pairs. Using only standard contrastive loss and no pseudo-labeling, our approach is a simple yet effective extension of the contrastive learning method to learn emergent object-semantic cues. In addition, we propose a shifted-window learning approach upon window attention to make the backbone representation more robust, translation-invariant, and less biased by the window pattern. On the popular LVIS open-vocabulary detection benchmark, our approach sets a new state of the art of 40.4 mask $AP_r$ using the common ViT-L backbone, significantly outperforming the best existing approach by +6.5 mask $AP_r$ at system level. On the COCO benchmark, we achieve very competitive 40.8 novel AP without pseudo labeling or weak supervision. In addition, we evaluate our approach on the transfer detection setup, where ours outperforms the baseline significantly. Visualization reveals emerging object locality from the pretraining recipes compared to the baseline. Code and models will be publicly released.

## 1 INTRODUCTION

To understand and localize all objects and entities in the visual world has been a foundational problem in computer vision and machine learning. This capability unlocks a broad array of compelling applications from self-driving cars to search and recommendation. However, existing object detectors typically rely on human-annotated regions and class labels. These annotations are costly and unscalable in terms of the number of categories *e.g.* O(1K) and the number of images *e.g.* O(100K).

The open-vocabulary detection (OVD) task (Zareian et al., 2021) has been introduced to overcome both limitations by pretraining on larger-scale image-text data before finetuning for detection tasks. By representing each category as a text embedding rather than a discrete label, open-vocabulary detectors can localize objects based on any user-provided text queries unavailable during training. Many techniques such as knowledge distillation (Gu et al., 2022; Du et al., 2022), weak supervision (Zhou et al., 2022b), self-training (Zhong et al., 2022; Rasheed et al., 2022; Zhao et al., 2022; Huynh et al., 2022), frozen backbone (Kuo et al., 2023), detection feature alignment (Arandjelović et al., 2023), and better positional embeddings (Kim et al., 2023b) have been proposed. Most existing methods assume the pretrained Vision Language Models (VLMs) *e.g.* (Radford et al., 2021) are given, and focus on recipes to train the entire model or at least the detector heads from scratch using the pretrained backbone (Gu et al., 2022; Du et al., 2022; Zhong et al., 2022; Kuo et al., 2023; Kim et al., 2023b; Rasheed et al., 2022). This tends to result in suboptimal generalization because the detector heads are trained on the limited vocabulary of detection datasets, and only the backbone contains the knowledge of open-vocabulary concepts.

We present DITO (Detection-Oriented Image-Text pretraining for Open-vocabulary detection), an intuitive method to perform image-language pretraining in a detection-friendly manner for open-vocabulary object detection. Standard pretraining typically uses image classification architecture, and thus needs to train at least the detector heads from scratch during the detection fine-tuning stage. In contrast, our approach learns the detector heads directly from the large image-text corpus. The detector heads receive text supervision at image level at multiple scales by pooling over randomly

generated regions, which learns image-text representation across multiple levels of semantic granularity. In addition, we proposed a simple yet effective Shifted-Window Learning (SWL) approach in detection to enhance the features of vision transformer using window attention. Specifically, we roll the patch tokens with strides less than the window size to mitigate the bias caused by window attention pattern and obtain a more shift-invariant representation. Through Detection-Oriented Pretraining (DOP) and Shifted-Window Learning (SWL), we close the gap between image-text pretraining and open-vocabulary detection, and obtain robust backbone features for better generalization.

The best DITO model obtains 40.4 mask $AP_r$ on the widely used LVIS open-vocabulary detection benchmark, surpassing the previous best approach by +6.5 $AP_r$ at system level. In the setting with external box annotations, it achieves 45.8 box $AP_r$, a significant gain of +12.5 points over the previous best approach. On the COCO benchmark, DITO achieves a very competitive 40.8 novel AP without using pseudo-labels or joint training. In summary, our contributions are:

- We present a novel contrastive pretraining methodology using detector architecture to learn detection-sensitive representation from noisy, large-scale image-text pairs.

- We propose a simple Shifted-Window Learning technique for detection to produce more robust and translation-invariant representation from pretrained vision transformers.

- Our approach significantly outperforms the state-of-the-art methods on LVIS open-vocabulary detection benchmark, including larger models and pseudo-labeling approaches, and achieves very competitive performance on COCO benchmark and transfer detection to Objects365 simultaneously.

We hope these findings will encourage the community to further explore detection-oriented pretraining on noisy, large-scale image-text data to advance open-vocabulary tasks.

## 2 RELATED WORK

**Language-supervised open-vocabulary recognition.** Learning representations for open-vocabulary recognition is a key to advancing general intelligence. To harness the rich co-occurrence pattern of images and text found in web data, researchers have explored a diverse range of data sources, including image tags, captions, alt-texts, image search queries, or combination thereof (Desai & Johnson, 2021; Sariyildiz et al., 2020; Radford et al., 2021; Jia et al., 2021; Schuhmann et al., 2021; Gadre et al., 2023; Chen et al., 2023). From a modeling standpoint, contrastive learning operates at the image level and has proven successful due to its simplicity, scalability, and versatility across zero-shot (Radford et al., 2021), linear probing (Radford et al., 2021), few-shot (Zhou et al., 2022a), and full finetuning (Dong et al., 2022) scenarios. Building upon this body of work, our approach learns region-aligned representations for open-vocabulary detection by leveraging detector architecture during the contrastive pretraining.

**Self-supervised pretraining for visual tasks.** Self-supervised learning has emerged as a promising paradigm to learn object features for complex visual tasks such as detection, given the challenge of scaling up human annotations. Early efforts designed pretext tasks that require semantic understanding to solve (Doersch et al., 2015; Pathak et al., 2016; Zhang et al., 2016). Subsequently, the idea of contrastive learning became popular where the pretext tasks are specified in the data itself, where the contrastive samples can take the forms of augmented images (Chen et al., 2020), sliding windows (Xiao et al., 2021), object proposals (Wei et al., 2021), or point samples (Bai et al., 2022). In addition to contrastive learning, alternative strategies like pseudo-labeling (Zhong et al., 2021), raw pixel (He et al., 2022) and image feature reconstruction (Bao et al., 2021; Zhou et al., 2021) have also proven effective. While the majority of these methods have focused on learning from images without textual context, and applying to closed-vocabulary detection, our work leverages large image-text data to tackle the more demanding open-vocabulary detection task.

**Open-vocabulary object detection and segmentation.** Open-Vocabulary detection has made very rapid progress in recent years (Zareian et al., 2021; Gu et al., 2022; Zhong et al., 2022). A common approach is to repurpose the capabilities of pretrained vision-language models (VLM) for detection. Various techniques including knowledge distillation (Gu et al., 2022) and prompt optimization (Du et al., 2022) have been used to train an open-vocabulary detector with the pretrained VLM. Pseudo-boxes and weak-labeling methods (Zhong et al., 2022; Li et al., 2022a; Zhou et al.,

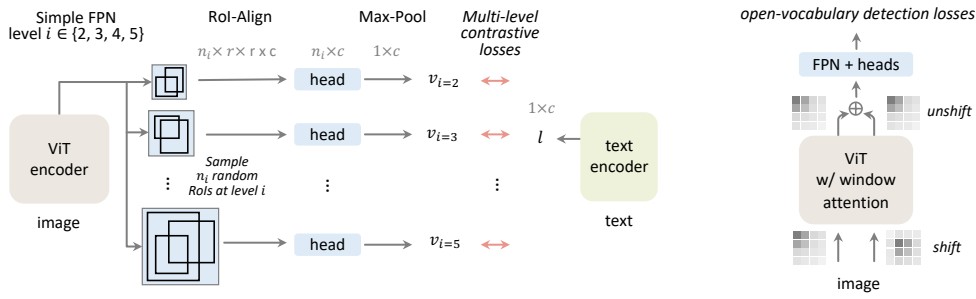

Figure 1: **DITO method.** Detection-Oriented Pretraining (left): DITO trains the detector heads (e.g. FPN (Li et al., 2022b; Lin et al., 2017a), Faster RCNN head (Ren et al., 2015)) upon a ViT encoder backbone with multi-level image-text contrastive loss to bridge the gap between image-text pretraining and open-vocabulary detection. Shifted-Window Learning for detection (right): DITO rolls the image and combine the shifted features with the original features to mitigate the bias of window attention grid (Li et al., 2022b), and produce more robust semantic representation.

2022b; Gao et al., 2022; Feng et al., 2022) have also been used to recover the region-level information missing from the image-level pretraining. In addition, pretrained VLM backbone can be directly employed by adding new detection heads either by setting the backbone frozen (Kuo et al., 2023) or finetunable (Minderer et al., 2022; Kim et al., 2023b).

Pretraining the vision-language models for open-vocabulary object detection is a recent direction. Yao et al. (2023); Zhang et al. (2022) train on a combination of a detection, grounding, and caption data to learn the word-region alignment. RO-ViT (Kim et al., 2023b) proposes region-aware positional embeddings for contrastive model training and uses the focal loss (Lin et al., 2017b) for contrastive learning. Our work is more closely related to the latter approaches, where object detection-oriented pretrainig is built in the pretraining phase of our model. Regarding backbone architecture, ConvNet, ViT (Dosovitskiy et al., 2020) or hybrid models (Liu et al., 2021) have been used in existing works. We adopt ViT with window attention in this work, and propose a shifted window learning approach to mitigate the window-induced bias.

## 3 METHOD

We address the problem of open-vocabulary object detection. At training time, the model can access the class and box labels of base categories ($C_B$). At test time, the model is tasked with detecting objects from a set of novel categories ($C_N$) not present in the training set. To achieve this, we leverage pretrained vision and language models (VLMs) building upon prior studies (Gu et al., 2022; Zhong et al., 2022; Kuo et al., 2023). However, instead of taking classification-based pretrained VLMs, we demonstrate how to enhance the VLMs with detector head in image-text pretraining, and propose shifted-window learning (SWL) strategy to improve open-vocabulary detection.

### 3.1 PRELIMINARIES

**Image-text pretraining.** Inspired by prior works in open-vocabulary detection, we adopt dual-encoder contrastive image-language pretraining (Radford et al., 2021; Jia et al., 2021) before applying the detection finetuning. The image embeddings $\{v\}$ and text embeddings $\{l\}$ are the average-pooled outputs from the image and text encoders, respectively. As in previous works, we compute the dot product of the embeddings in batch $B$, and scale it by a learnable temperature $\tau$ before applying the InfoNCE loss (Oord et al., 2018; Radford et al., 2021). Mathematically, the image-to-text (I2T) loss can be expressed as:

$$L_{\text{I2T}} = -\frac{1}{B} \sum_{i=1}^{B} \log\left(\frac{\exp(v_i l_i / \tau)}{\sum_{j=1}^{B} \exp(v_i l_j / \tau)}\right). \tag{1}$$

The text-to-image (T2I) loss is symmetrical by exchanging the inner/outer summation loops. The total contrastive loss $L_{con}$ is obtained by $L_{con} = (L_{\text{I2T}} + L_{\text{T2I}})/2$.

**Open-vocabulary detection finetuning.** At the fine-tuning stage, our detection finetuning recipe follows previous studies (Zareian et al., 2021; Gu et al., 2022; Kuo et al., 2023; Kim et al., 2023b). During the training phase, we use the RoI-Align (He et al., 2017) feature as the detection embedding for each detected region. We replace the fixed-size classifier layer with the text embeddings of base categories ($C_B$). The detection score $p_i$ is determined by calculating the cosine similarity between the region embedding $r_i$ and text embeddings of base categories ($C_B$) followed by a softmax operation. We prepend an additional background class embedding to $C_B$ and use the term "background" to represent the background category. Any proposals that do not match to any base category annotations are treated as background during training. It is important that the text embeddings are computed from the same text encoder as from the image-text pretraining. During testing, we expand the text embeddings to include the novel categories ($C_B \cup C_N$), resulting in ($C_B \cup C_N + 1$) categories including the background. We calculate the detection scores ($p_i$) as the cosine similarity between the region embeddings ($r_i$) and the expanded text embeddings.

Apart from the detection embedding ($r_i$), we extract the VLM embedding (Kuo et al., 2023) of region $i$ by RoI-Align at the last ViT backbone feature map. The VLM score ($z_i$) is calculated as the cosine similarity with the text embeddings of the combined categories ($C_B \cup C_N$). As we train the ViT backbone for detection tasks, there is a tendency to lose the pretrained image-text knowledge. Inspired by previous work (Kim et al., 2023a), we use a separate frozen ViT backbone as an open-vocabulary region classifier at inference time to compute the VLM score $z_i$.

To compute the open-vocabulary detection score ($s_i^{\text{ens}}$), we ensemble the detection and VLM scores by geometric means (Gu et al., 2022; Kuo et al., 2023). The formula is as follows:

$$s_i^{\text{ens}} = \begin{cases} z_i^{(1-\alpha)} \cdot p_i^{\alpha} & \text{if } i \in C_B \\ z_i^{(1-\beta)} \cdot p_i^{\beta} & \text{if } i \in C_N \end{cases} \tag{2}$$

Here, $\alpha, \beta$ are floats $\in [0, 1]$ that control the weighting of base versus novel categories. The background score comes from the detection score ($p_i$) alone, because we observe the VLM score of "background" class is often less reliable.

## 3.2 DETECTION-ORIENTED PRETRAINING

Standard image-text pretraining uses classification architectures because the language supervision occurs at the image level rather than object level. Despite showing strong zero-shot classification performance, this approach often leads to sub-optimal performance for detection tasks. To bridge this gap, we propose Detection-Oriented Pretraining (DOP) to use detection architecture instead to capture the rich region-level learning signal in the pretraining phase through the object-oriented design of detection models such as FPN (Lin et al., 2017a; Li et al., 2022b), RoI-Align (He et al., 2017). In addition, pretraining with detection architecture allows us to transfer not only the weights of the image backbone, but also the weights of the detector heads to downstream finetuning tasks. Thus, the detector heads are not trained from scratch on a limited set of categories, but warm-started from the knowledge of large image-text data, thereby improving the generalization capability.

**Detector head learning from random regions.** Figure 1 (left) illustrates our detection-oriented pretraining system. To achieve the goal of training detector heads on image-text data, we first attach the simple feature pyramid network (Li et al., 2022b) on the output features of vision transformer backbone. This is because the vision transformer backbone does not provide multi-level feature maps as needed by the standard feature pyramid (Lin et al., 2017a). On top of the feature pyramid, we apply the RoI-Align (He et al., 2017) and Faster R-CNN (Ren et al., 2015) classifier head to match the classification pathway of pretraining with the region-recognition pathway in detection finetuning (see Table 5a for ablations).

For each level $i$ of the feature pyramid, we randomly generate $n_i$ box regions uniformly over the image by sampling the box size $h, w \sim Uniform(0.2, 1.0)$ and aspect ratio $h/w \sim Uniform(0.5, 2.0)$. The $n_i$ value is set proportional to the size of the $i$-th feature map so that larger feature map would be covered by more regions. We extract the RoI-features of each region by RoI-Align, and feed them through the region classifier head (Ren et al., 2015) to obtain the RoI embedding.

**Multi-level image-text supervision.** After computing the RoI embeddings across pyramid levels, we perform a max pooling over the RoI embeddings per-level to obtain an image embedding for each pyramid level. Intuitively, max pooling allows the representation to focus on salient regions and

discriminative features, thereby learning region-level information without explicit supervision. Then we apply the standard image-text contrastive loss (see Eqn. 1) on each feature level separately, which aids the learning of rich semantic and object knowledge within every feature map (see Table 5b for ablations). The losses from all levels are weighted equally and summed together. As a result of the multi-level learning, we observe the feature maps possess more localized semantic information compared to the baseline (see Figure 2).

Different from pseudo-labeling techniques (Zhong et al., 2022; Feng et al., 2022; Huynh et al., 2022) that require additional steps to prepare and store annotations, our approach learns the detector heads on the fly via contrastive learning without a need to cache annotations or detection-specific losses.

### 3.3 SHIFTED-WINDOW LEARNING FOR DETECTION

We perform shifted-window learning on top of the vanilla vision transformer (Dosovitskiy et al., 2020) without adding or removing any weights. In order to run vision transformer on large detection images (e.g. $1024 \times 1024$), we apply window-attention layers on a fixed-size grid $K \times K$, and $L$ global attention layers evenly spaced throughout the vision transformer (where $L = 4$) following Li et al. (2022b). Although information mixing occurs $L$ times through global attention, we observed that fixed-size grid is still biased by the grid pattern, thereby compromising the representation power of the backbone. To address the issue, we propose the Shifted-Window Learning (SWL) approach to mitigate the bias of grid pattern in window-attention vision transformer.

**Network Architecture.** Figure 1 (right) illustrates our shifted-window learning system. The standard vision transformer consists of a patchifying layer and a set of transformer layers. After feeding the image through the patchifying layer, we obtain a feature map of shape $(h, w, c)$. We keep a copy of this feature map to feed through the rest of the ViT, and create another copy which we roll along both $h$ and $w$ axes by $s$ pixels. The elements that roll beyond the last position are reintroduced from the first. The shift size $s$ is set as a fraction $q$ of the window attention cell size $M$, i.e. $s = qM$. Empirically, we set $q = 0.5$ and the cell size $M = 16$ equals the image size (e.g. 1024) divided by the product of patch size (e.g. 16) and the grid size (e.g. 4). We feed the shifted feature map through the rest of the ViT in the same way as the original feature map. The outputs are two sequence features of the same shape $(h, w, c)$. We then unshift the shifted features before combining the two sequences by averaging. We apply the above shifted window operations in both detection training and test times (see Table 6 for ablations on various configurations), and observe clear improvements in representation quality.

**Comparison with other architectures.** Compared to the Swin Transformer (Liu et al., 2021), we apply the shifted-window ideas as separate forward passes, while Swin Transformer applies similar ideas in an alternating manner layer by layer. Our approach requires no change to the vanilla transformer architecture and is compatible with any ViT backbones pretrained without shifted windows (e.g. (Radford et al., 2021)), whereas Swin Transformer requires specialized pretraining on the same architecture. Compared to the full-attention vision transformer (Dosovitskiy et al., 2020), we observe that window attention taps more effectively into the region-level semantic knowledge of pretrained backbone than full attention, perhaps because the window structure helps the model focus on relevant local cues and be more robust to noises farther away.

## 4 EXPERIMENTAL RESULTS

**Pretraining setup.** Our image-text pretraining consists of two phases. We first train a contrastive VLM from scratch following the standard CLIP (Radford et al., 2021) recipe for 500K iterations, with 16k batch size, and 224×224 image size. We use the vision transformers as image encoders and use global average pooling at the last layer instead of CLS-token pooling.

Next, we apply the detection-oriented pretraining (DOP) where we freeze the image and text encoders trained in the first phase and introduce the detector heads. We use the Simple FPN (Li et al., 2022b) and Faster R-CNN classifier head, where we replace the batch normalization with layer normalization. At the $i$-th pyramid level $i \in \{2, 3, 4, 5\}$, we randomly sample $n_i \in \{400, 200, 100, 50\}$ box regions and compute their RoI-Align features. We use a short training cycle of 30K iterations, 4k batch size, 256×256 image size, AdamW optimizer with an initial learning rate (LR) of 1e-4, linear LR decay and warmup of 5k iterations. The ALIGN (Jia et al., 2021) image-text dataset is used by default, although we also explore the more recent DataComp-1B (Gadre et al., 2023). To

Table 1: **LVIS open-vocabulary detection (mask/box AP).** DITO outperforms the best existing approach by +6.5 mask $AP_r$ in the standard benchmark, and by +12.5 box $AP_r$ in the unconstrained setting (Minderer et al., 2022). ⋆: uses the public DataComp-1B (Gadre et al., 2023) data in pretraining. §: uses filtered Objects365 box annotations in detection training.

| method | pretrained model | detector backbone | **mask $AP_r$** | mask AP | **box $AP_r$** | box AP |
|---|---|---|---|---|---|---|
| *ConvNet based:* | | | | | | |
| VL-PLM (Zhao et al., 2022) | ViT-B/32 | R-50 | 17.2 | 27.0 | - | - |
| OV-DETR (Zang et al., 2022) | ViT-B/32 | R-50 | 17.4 | 26.6 | - | - |
| DetPro-Cascade (Du et al., 2022) | ViT-B/32 | R-50 | 20.0 | 27.0 | 21.7 | 30.5 |
| Rasheed (Rasheed et al., 2022) | ViT-B/32 | R-50 | 21.1 | 25.9 | - | - |
| PromptDet (Feng et al., 2022) | ViT-B/32 | R-50 | 21.4 | 25.3 | - | - |
| OADB (Wang et al., 2023) | ViT-B/32 | R-50 | 21.7 | 26.6 | 21.9 | 28.7 |
| RegionCLIP (Zhong et al., 2022) | R-50x4 | R-50x4 | 22.0 | 32.3 | - | - |
| CORA (Wu et al., 2023b) | R-50x4 | R-50x4 | - | - | 22.2 | - |
| BARON (Wu et al., 2023a) | ViT-B/32 | R-50 | 22.6 | 27.6 | 23.2 | 29.5 |
| CondHead (Wang, 2023) | R-50x4 | R-50x4 | 24.4 | 32.0 | 25.1 | 33.7 |
| Detic-CN2 (Zhou et al., 2022b) | ViT-B/32 | R-50 | 24.6 | 32.4 | - | - |
| ViLD-Ens (Gu et al., 2022) | EffNet-B7 | EffNet-B7 | 26.3 | 29.3 | 27.0 | 31.8 |
| 3Ways(Arandjelović et al., 2023) | NFNet-F6 | NFNet-F6 | - | - | 30.1 | 44.6 |
| F-VLM (Kuo et al., 2023) | R-50x64 | R-50x64 | 32.8 | 34.9 | - | - |
| *ViT based:* | | | | | | |
| OWL-ViT (Minderer et al., 2022) | ViT-H/14 | ViT-H/14 | - | - | 23.3 | 35.3 |
| OWL-ViT (Minderer et al., 2022) | ViT-L/14 | ViT-L/14 | - | - | 25.6 | 34.7 |
| RO-ViT (Kim et al., 2023b) | ViT-B/16 | ViT-B/16 | 28.0 | 30.2 | 28.4 | 31.9 |
| RO-ViT (Kim et al., 2023b) | ViT-L/16 | ViT-L/16 | 32.1 | 34.0 | 33.6 | 36.2 |
| CFM-ViT (Kim et al., 2023a) | ViT-B/16 | ViT-B/16 | 28.8 | 32.0 | 29.6 | 33.8 |
| CFM-ViT (Kim et al., 2023a) | ViT-L/16 | ViT-L/16 | 33.9 | 36.6 | 35.6 | 38.5 |
| **DITO (ours)** | ViT-B/16 | ViT-B/16 | **32.5** | 34.0 | **34.9** | 36.9 |
| **DITO (ours)** | ViT-L/16 | ViT-L/16 | **38.4** | 36.3 | **41.1** | 40.4 |
| **DITO (ours)** ⋆ | ViT-L/16 | ViT-L/16 | **40.4** | 37.7 | **42.8** | 40.7 |
| *with external box annotations:* | | | | | | |
| OWL-ViT (Minderer et al., 2022) § | ViT-B/16 | ViT-B/16 | - | - | 20.6 | 27.2 |
| OWL-ViT (Minderer et al., 2022) § | ViT-L/14 | ViT-L/14 | - | - | 31.2 | 34.6 |
| DetCLIPv2 (Yao et al., 2023) § | Swin-L | Swin-L | - | - | 33.3 | 36.6 |
| **DITO (ours)** § | ViT-L/16 | ViT-L/16 | - | - | **45.8** | 44.2 |

improve the region-awareness of the backbone, we adopt the cropped positional embedding (Kim et al., 2023b) for both phases of pretraining.

**Detection finetuning setup.** We train the detector with image size $1024 \times 1024$ and use window attention in the backbone with grid size 4×4 as in Li et al. (2022b). The learning rate for the backbone is set lower as $0.6\times$ of the detector head layers. The text embedding of each category is calculated as the average over the CLIP prompt templates. We use the batch size 128, the SGD optimizer with momentum 0.9. The initial learning rate and iterations are set to 0.18 and 36.8k for LVIS, and 0.02 and 11.3k for COCO datasets.

## 4.1 COMPARISON TO THE STATE-OF-THE-ART

**Comparisons on LVIS.** In Table 1, we report the comparison with existing methods on the challenging LVIS benchmark. The 'frequent' and 'common' classes of the dataset belong to the base categories $C_B$, and the 'rare' classes are the novel categories $C_N$. The primary benchmark metric is the mask AP on rare classes (mask $AP_r$). The best DITO model achieves the performance of 38.4 mask $AP_r$, which significantly outperforms the current state-of-the-art approaches RO-ViT and CFM-ViT with the same ViT-L backbone by +6.3 and +4.5 points using the same pretraining data (Jia et al., 2021). We achieve 40.4 mask $AP_r$ when we replace the ALIGN (Jia et al., 2021) with DataComp-1B (Gadre et al., 2023). With the smaller ViT-B backbone, DITO maintains a healthy margin of around +4 $AP_r$ above existing approaches based on ViT-B.

In addition, we present system-level comparisons in the unconstrained setting of Minderer et al. (2022) where additional non-LVIS$_{rare}$ object annotations *e.g.* Objects365 (Shao et al., 2019) can be used in detection training (denoted by § in Table 1), and report the box $AP_r$ metric. In this setting, we

Table 2: **COCO open-vocabulary detection (box AP50).** DITO demonstrates a very competitive novel category AP without using pseudo labeling or weak supervision. ‡: uses an external MViT detector (Maaz et al., 2022). §: uses filtered Objects365 annotations for training.

| method | pretrained model | detector backbone | **novel AP** | AP |
|---|---|---|---|---|
| *ConvNet based:* | | | | |
| OV-RCNN (Zareian et al., 2021) | R-50 | R-50 | 22.8 | 39.9 |
| ViLD (Gu et al., 2022) | ViT-B/32 | R-50 | 27.6 | 51.3 |
| F-VLM (Kuo et al., 2023) | R-50 | R-50 | 28.0 | 39.6 |
| OV-DETR (Zang et al., 2022) | ViT-B/32 | R-50 | 29.4 | 52.7 |
| **with pseudo box labels:** | | | | |
| PromptDet (Feng et al., 2022) | ViT-B/32 | R-50 | 26.6 | 50.6 |
| XPM (Huynh et al., 2022) | R-50 | R-50 | 27.0 | 41.2 |
| OADB (Wang et al., 2023) | ViT-B/32 | R-50 | 30.0 | 47.2 |
| CondHead (Wang, 2023) | R-50 | R-50 | 33.7 | 51.7 |
| VL-PLM (Zhao et al., 2022) | ViT-B/32 | R-50 | 34.4 | 53.5 |
| RegionCLIP (Zhong et al., 2022) | R-50x4 | R-50x4 | 39.3 | 55.7 |
| CORA (Wu et al., 2023b) | R-50x4 | R-50x4 | **43.1** | 56.2 |
| **with weak supervision:** | | | | |
| Detic-CN2 (Zhou et al., 2022b) | ViT-B/32 | R-50 | 24.6 | 32.4 |
| *ViT based:* | | | | |
| RO-ViT (Kim et al., 2023b) | ViT-B/16 | ViT-B/16 | 30.2 | 41.5 |
| RO-ViT (Kim et al., 2023b) | ViT-L/16 | ViT-L/16 | 33.0 | 47.7 |
| CFM-ViT (Kim et al., 2023a) | ViT-B/16 | ViT-B/16 | 30.8 | 42.4 |
| CFM-ViT (Kim et al., 2023a) | ViT-L/16 | ViT-L/16 | 34.1 | 46.0 |
| **DITO (ours)** | ViT-B/16 | ViT-B/16 | **38.6** | 48.5 |
| **DITO (ours)** | ViT-L/16 | ViT-L/16 | **40.8** | 50.3 |
| *methods with external box annotations:* | | | | |
| Rasheed (Rasheed et al., 2022) ‡ | ViT-B/32 | R-50 | 36.9 | 51.5 |
| BARON (Wu et al., 2023a) ‡ | R-50 | R-50 | 42.7 | 51.7 |
| **DITO (ours)** § | ViT-L/16 | ViT-L/16 | **46.1** | 54.2 |

Table 3: **Transfer detection from LVIS to Objects365 (box AP).** All models are tested on Objects365 dataset following the setup of Gu et al. (2022). DITO outperforms existing methods with comparable backbone size.

| method | backbone | AP | $AP_{50}$ | $AP_{75}$ |
|---|---|---|---|---|
| Supervised (Gu et al., 2022) | R-50 | 25.6 | 38.6 | 28.0 |
| ViLD (Gu et al., 2022) | R-50 | 11.8 | 18.2 | 12.6 |
| DetPro (Du et al., 2022) | R-50 | 12.1 | 18.8 | 12.9 |
| BARON (Wu et al., 2023a) | R-50 | 13.6 | 21.0 | 14.5 |
| F-VLM (Kuo et al., 2023) | R-50x16 | 16.2 | 25.3 | 17.5 |
| F-VLM (Kuo et al., 2023) | R-50x64 | 17.7 | 27.4 | 19.1 |
| RO-ViT (Kim et al., 2023b) | ViT-L/16 | 17.1 | 26.9 | 18.5 |
| CFM-ViT (Kim et al., 2023a) | ViT-L/16 | 18.7 | 28.9 | 20.3 |
| **DITO (ours)** | ViT-L/16 | **19.8** | **30.4** | **21.2** |

additionally pretrain the detector on Objects365 annotations with $LVIS_{rare}$ categories filtered out. Then, we conduct the same $LVIS_{base}$ training in the standard setting. The additional Objects365 annotations provide a significant boost to DITO and yields the state-of-the-art 45.8 box $AP_r$, +12.5 $AP_r$ over the state of the art in this setting.

**Comparisons on COCO.** We present the comparisons on COCO benchmark in Table 2. The main metric is AP50 of novel categories (novel AP). Our model demonstrates a very competitive result 40.8 novel AP without using pseudo labeling (Feng et al., 2022; Huynh et al., 2022; Zhao et al., 2022; Zhong et al., 2022), weak supervision (Zhou et al., 2022b), or externally trained detector modules (Rasheed et al., 2022; Wu et al., 2023a). Among the ViT-based methods, DITO outperforms RO-ViT (Kim et al., 2023b) and CFM-ViT (Kim et al., 2023a) by a clear margin of +7.2 and +6.1 points, respectively. In the unconstrained setting, we observe that training on additional box labels of Shao et al. (2019) before the $COCO_{base}$ training further improves DITO to 46.1 novel AP, surpassing all existing works that use external box annotations. The Objects365 annotations are deduplicated against all 80 COCO categories.

Table 4: **Ablation studies** on LVIS benchmark. We train on base ('frequent' + 'common') categories, and report $AP_r$ on novel ('rare') categories

(a) **Detection-Oriented Pretraining (DOP) and Shifted-Window Learning (SWL)**.

| pretraining method | $AP_r$ | AP |
|---|---|---|
| baseline | 36.2 | 37.4 |
| w/ DOP | 38.5 (+2.3) | 38.3 |
| w/ SWL | 40.3 (+4.1) | 39.7 |
| w/ DOP + SWL | 41.2 (+5.0) | 40.5 |

(b) **Frozen backbone inference**.

| detector inference | $AP_r$ | AP |
|---|---|---|
| baseline | 36.2 | 37.4 |
| *without* frozen backbone | 32.8 (-3.4) | 36.3 |

Table 5: **Ablation on Detection-Oriented Pretraining (DOP).** Best setting is marked by gray.

(a) **Detector components**.

| pretraining method | $AP_r$ | AP |
|---|---|---|
| baseline | 36.2 | 37.4 |
| w/ FPN | 37.2 (+1.0) | 37.5 |
| w/ FPN + Head | 38.5 (+2.3) | 38.3 |

(b) **RoI sampling and pooling**.

| RoI embedding | $AP_r$ | AP |
|---|---|---|
| single whole-image RoI | 37.8 | 38.0 |
| multiple RoIs, mean per level | 37.5 | 37.9 |
| multiple RoIs, max per level | 38.5 | 38.3 |
| multiple RoIs, max all levels | 37.7 | 38.1 |

Table 6: **Ablation on Shifted-Window Learning (SWL).** Our chosen setting is marked by gray.

(a) **Shifted-window backbone**.

| shifted-window | $AP_r$ | AP |
|---|---|---|
| baseline | 36.2 | 37.4 |
| frozen backbone | 37.4 (+1.2) | 37.6 |
| finetuned backbone | 38.9 (+2.7) | 39.4 |
| both backbones | 40.3 (+4.1) | 39.7 |

(b) **Window division**.

| grid | $AP_r$ | AP |
|---|---|---|
| 1×1 | 35.7 | 37.6 |
| 2×2 | 37.7 | 39.0 |
| 4×4 | 40.3 | 39.7 |
| 8×8 | 35.2 | 37.9 |

(c) **Window shift size**.

| stride | $AP_r$ | AP |
|---|---|---|
| 16 | 36.2 | 37.4 |
| 8 | 40.3 | 39.7 |
| 4 | 40.6 | 39.9 |

**Transfer detection.** We further evaluate DITO in the transfer detection setting, where the open-vocabulary detector trained on one dataset is tested on another dataset without any finetuning. Following the setup of ViLD (Gu et al., 2022), the detectors trained on $LVIS_{base}$ are evaluated on Objects365 dataset by simply replacing the text embeddings. Table 3 shows that DITO achieves 19.8 AP, outperforming previous methods using ConvNet or ViT backbones of similar size.

## 4.2 ABLATION STUDIES

To study the advantages of DITO methods, we provide ablation studies on the LVIS benchmark. We use ViT-L/16 backbone and report box $AP_r$ for ablations.

**DITO framework.** In Table 4a, we verify that the gains demonstrated by DITO come from our proposed detection-oriented pretraining (DOP) as well as the shifted-window learning (SWL). Our baseline uses the CLIP pretraining with cropped positional embedding (Kim et al., 2023b). For detection, the baseline adopts the frozen backbone inference method as in CFM-ViT (Kim et al., 2023a). It already achieves a very strong performance of 36.2 $AP_r$, higher than the state-of-the-art CFM-ViT by +0.6 point. On top of this, our proposed detection-oriented pretraining achieves a gain of +2.3 $AP_r$ and SWL improves the baseline by +4.1 $AP_r$. By combining both strategies, DITO shows a significant gain of +5.0 $AP_r$ over the baseline. Table 4b reports that excluding the frozen backbone inference leads to a drop of -3.4 $AP_r$, which is consistent with the findings in CFM-ViT.

**Detection-Oriented Pretraining.** To further investigate our detection-oriented image-text pretraining, we ablate in Table 5a by progressively adding the FPN and Faster R-CNN head into the pretraining. The 'w/ FPN' only introduces the FPN, where each pyramid level map is mean-pooled into an image embedding, followed by the image-text contrastive loss per level. It improves the baseline by +1.0 $AP_r$. Incorporating both FPN and Faster R-CNN head (see Section 3.2) further improves the alignment between the pretraining and the detection finetuning, achieving 38.5 $AP_r$, a gain of +2.3 points over the baseline. Table 5b ablates different RoI sampling strategies and the pooling of the RoI embeddings. We first observe that sampling multiple RoIs instead of using the whole-image RoI achieves the best performance. After computing the RoI embeddings, we find max pooling over the RoI embeddings performs better than mean pooling, and pooling per pyramid level (*i.e.*, multi-level image-text supervision) is better than pooling over all levels.

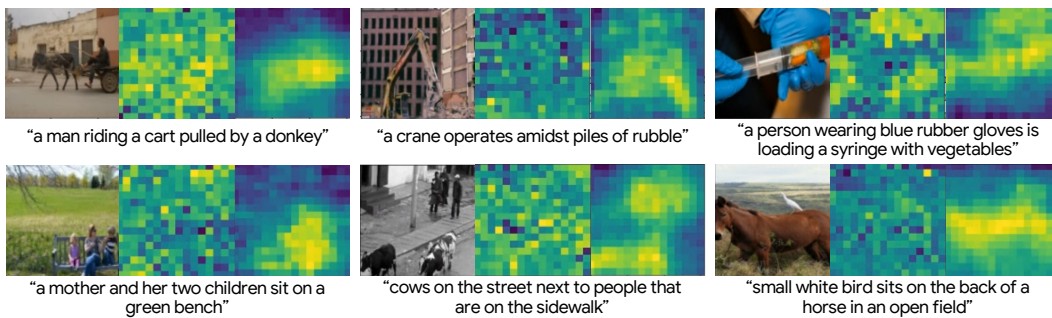

Figure 2: **Visual-text similarity map**. For each example, we show the paired image (left) and text (bottom) input, and the visual-text similarity map using the backbone features (middle) or our detection-oriented pretraining features (right). We use Flickr30K (top row) and COCO Captions (bottom row) datasets.

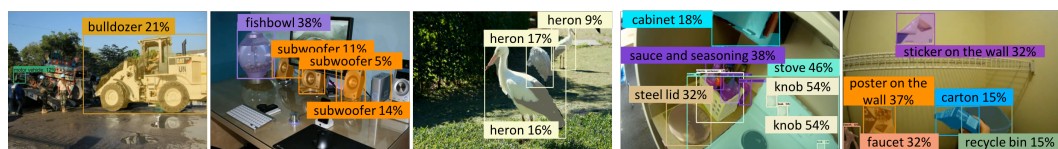

Figure 3: **DITO prediction.** LVIS results (left three) only show the novel categories (*e.g.*, *bulldozer, fishbowl, subwoofer, heron*). While Ego4D (right two) is a real-world and out-of-distribution data, many unseen objects are detected (*e.g.*, *steel lid, sticker on the wall, recycle bin*). Best viewed with zoom in.

**Shifted-Window Learning for detection.** As discussed in Section 3.1, our baseline detector adopts a separate frozen ViT backbone at inference (Kim et al., 2023a) to compute the VLM scores $p_i$ in Equation 2. In Table 6a, we ablate applying shifted window in the frozen and finetuned (detector) backbones. We find that it improves the performance in all cases, and using the shifted window in both backbones brings the largest gain of +4.1 $AP_r$. Table 6b compares different window attention division. Note that the grid size '1×1' uses global attention throughout all layers without any window attention and thus without shifting operation. OWL-ViT (Minderer et al., 2022) uses this full global attention with its pretrained CLIP ViT backbone. Interestingly, we observe that our SWL outperforms the full global attention by +4.6 $AP_r$ with the grid size 4×4 being the optimal value. Table 6c ablates the window shift size ($s$ in section 3.3). Setting the shift size the same as the window attention cell size (*i.e.*, 16) is equivalent to the non-shifted window baseline. We choose stride $s = 8$ in our method as a denser stride brings only marginal improvements.

### 4.3 VISUALIZATION

In Figure 2, we visualize the similarity map between the image features and a query text embedding using the Flickr30K (Plummer et al., 2015) and COCO Captions (Chen et al., 2015b) datasets. For each sample, we compare the baseline backbone features (middle) and our detection-oriented pretraining features (right). We select pyramid level 4 which has the same resolution as the backbone features and apply the Faster R-CNN head in a sliding window manner to obtain the dense feature map. We observe that our detection-oriented pretraining results in more localized semantic information given the queried image-text pairs. In Figure 3, we visualize the DITO outputs on LVIS novel categories and Ego4D (Grauman et al., 2022) which is real-world and out-of-distribution data. We use the same DITO detector trained on LVIS$_{base}$, and observe that it is able to capture many novel and unseen objects even under the significant domain shift.

### 5 CONCLUSION

We introduce DITO – a detection-oriented approach to learn from noisy image-text pairs for open-vocabulary detection. By replacing the classification architecture with detection architecture, DITO learns the locality-sensitive information from large-scale image-text data without a need for pseudo-labeling or additional losses. Furthermore, we present a shifted-window learning method to mitigate the bias of window attention pattern in vision transformer detectors. Experiments show that DITO outperforms the state-of-the-art by large margins on the LVIS benchmark, and is very competitive on the COCO benchmark and transfer detection. We hope this work would inspire the community to explore detection-oriented image-text pretraining for open-vocabulary localization tasks.

## 6 ETHICS STATEMENT

Our models utilize the rich image-text information acquired through pretraining, which may reinforce deficiencies and biases in the raw web data and expose potentially harmful biases or stereotypes. The models we trained are designed for academic research purposes and need more rigorous fairness studies before serving other applications.

## 7 REPRODUCIBILITY STATEMENT

We plan to open source the code for reproducibility. The model, experimental and implementation details are provided in the paper and supplementary materials. The detector model (He et al., 2017; Li et al., 2022b), pretraining image-text data (Gadre et al., 2023), and detection datasets (Gupta et al., 2019; Shao et al., 2019) in this work are publicly available.

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

# A APPENDIX

## A.1 LIGHTWEIGHT BACKBONE COMPARISON

Existing ViT-based open-vocabulary detection approaches typically focus on backbones heavier than the commonly used ResNet-50. In Table 7, we present additional LVIS benchmark results using a light-weight ViT-**S**/16 backbone, which is comparable in parameter count to ResNet-50. DITO outperforms the previous best approach using ResNet-50 backbone by a clear margin of +5.2 mask $AP_r$. We use DataComp-1B (Gadre et al., 2023) data for pretraining in this experiment.

## A.2 PRETRAINING AND FINETUNING PARAMETERS

Table 8 summarizes the hyperparameters used in our Detection-Oriented Pretraining (DOP) and open-vocabulary detection finetuning.

Table 7: **Lightweight backbone benchmark on LVIS.** DITO with ViT-**S** backbone significantly outperforms the best approach based on ResNet-50 by +5.2 mask $AP_r$.

| method | pretrained model | detector backbone | **mask $AP_r$** | mask AP | **box $AP_r$** | box AP |
|---|---|---|---|---|---|---|
| ViLD-ens (Gu et al., 2022) | ViT-B/32 | R-50 | 16.6 | 25.5 | | |
| VL-PLM (Zhao et al., 2022) | ViT-B/32 | R-50 | 17.2 | 27.0 | - | - |
| OV-DETR (Zang et al., 2022) | ViT-B/32 | R-50 | 17.4 | 26.6 | - | - |
| RegionCLIP (Zhong et al., 2022) | R-50 | R-50 | 17.4 | 26.7 | 17.1 | 28.2 |
| F-VLM (Kuo et al., 2023) | R-50 | R-50 | 18.6 | 24.2 | - | - |
| Detic (Zhou et al., 2022b) | ViT-B/32 | R-50 | 19.5 | 30.9 | - | - |
| DetPro-Cascade (Du et al., 2022) | ViT-B/32 | R-50 | 20.0 | 27.0 | 21.7 | 30.5 |
| Rasheed (Rasheed et al., 2022) | ViT-B/32 | R-50 | 21.1 | 25.9 | - | - |
| PromptDet (Feng et al., 2022) | ViT-B/32 | R-50 | 21.4 | 25.3 | - | - |
| OADP (Wang et al., 2023) | ViT-B/32 | R-50 | 21.7 | 26.6 | 21.9 | 28.7 |
| BARON (Wu et al., 2023a) | ViT-B/32 | R-50 | 22.6 | 27.6 | 23.2 | 29.5 |
| **DITO (ours)** | ViT-**S**/16 | ViT-**S**/16 | **27.8** | 29.4 | **29.0** | 31.3 |

Table 8: **Hyperparameters for Detection-Oriented Pretraining (left) and open-vocabulary detection fine-tuning (right)**

| Detection-Oriented Pretraining | | OVD finetuning | LVIS / COCO |
|---|---|---|---|
| optimizer | AdamW | optimizer | SGD |
| momentum | $\beta$=0.9 | momentum | $\beta$=0.9 |
| weight decay | 0.01 | weight decay | 0.0001 |
| learning rate | 0.0001 | learning rate | 0.18 / 0.02 |
| warmup steps | 5k | backbone lr ratio | 0.6× / 0.2× |
| total steps | 30k | step decay factor | 0.1× |
| batch size | 4096 | step decay schedule | [0.8, 0.9, 0.95] |
| image size | 256 | warmup steps | 1k |
| | | total steps | 36.8k / 11.3k |
| | | batch size | 128 |
| | | image size | 1024 |

## A.3 APPLICATION ON EGO4D DATA

In Figure 3, we test DITO's transfer detection to a real-world ego-centric data, Ego4D (Grauman et al., 2022). We use the same DITO trained on LVIS$_{base}$ categories with ViT-L/16 backbone

The categories are provided by the user based on visual inspection of the video as follows:

*plate, cabinet, stove, towel, cleaning rag, ventilator, knob, sauce and seasoning, steel lid, window, window blinds, plant, light switch, light, door, carpet, exit sign, doormat, hair, door lock, tree, poster on the wall, sticker on the wall, faucet, recycle bin, rack, hand, can, carton, trash, Christmas tree, plastic container, fridge.*

## A.4 DATASET LICENSE

- DataComp (Gadre et al., 2023): MIT License
- COCO (Lin et al., 2014): Creative Commons Attribution 4.0 License
- LVIS (Gupta et al., 2019): CC BY 4.0 + COCO license
- COCO Captions (Chen et al., 2015a): CC BY
- Flickr30k (Plummer et al., 2015): Custom (research-only, non-commercial)
- Objects365 (Shao et al., 2019): Custom (research-only, non-commercial)
- Ego4D (Grauman et al., 2022): https://ego4d-data.org/pdfs/Ego4D-Licenses-Draft.pdf

