# OpenReview forum: "Detection-Oriented Image-Text Pretraining for Open-Vocabulary Detection"
_ICLR.cc/2024/Conference — ICLR 2024 Conference Withdrawn Submission_

### Official Review · Reviewer_aquR · 2023-10-30

**Soundness:** 3 good
**Presentation:** 3 good
**Contribution:** 3 good
**Rating:** 5
**Confidence:** 3

**Summary:**

This paper targets detection-oriented image-text pertaining to the open-vocabulary detection task. The authors employ a detector architecture to facilitate contrastive learning from noisy, large-scale image-text pairs. Additionally, they introduce the Shifted-Window Learning technique to enhance detection performance and generate more robust, translation-invariant representations using pretrained vision transformers. Extensive experiments demonstrate the effectiveness of these methods.

**Strengths:**

- The paper is well-written and easy to follow.
- The motivation of the method is clear and the proposed strategy is straightforward yet effective.

**Weaknesses:**

My primary concern pertains to the novelty of the proposed method.  Firstly, the detection-oriented image-text pertaining has been explored extensively since RegionCLIP[1]. Subsequently, there have been endeavors to employ randomly selected proposals for augmentation [2] and align multiple regions [3]. Given the foundation laid by the aforementioned works, the contrastive pretraining methodology introduced in this article, employing a detector architecture, may appear to lack a sufficient level of novelty.
Secondly,  I think the shifted-window learning technique is not enough to support the second technical contribution. Overall, I agree with the practical values of this paper, but more scientific values from a paper in ICLR are expected.

[1] Yiwu Zhong, Jianwei Yang, Pengchuan Zhang, Chunyuan Li, Noel Codella, Liunian Harold Li, Luowei Zhou, Xiyang Dai, Lu Yuan, Yin Li, and Jianfeng Gao. Regionclip: Region-based languageimage pretraining. In CVPR, 2022.


[2] Wei, Fangyun, et al. "Aligning pretraining for detection via object-level contrastive learning." Advances in Neural Information Processing Systems 34 (2021): 22682-22694.


[3] Size Wu, Wenwei Zhang, Sheng Jin, Wentao Liu, and Chen Change Loy. Aligning bag of regions for open-vocabulary object detection. In Proceedings of the IEEE/CVF Conference on Computer Vision and Pattern Recognition, pp. 15254–15264, 2023a.

**Questions:**

Why didn’t the author compare the performance of Grounding-DINO? It has already had greater influence and zero-shot performance in the open-vocabulary detection field.

---

### Official Review · Reviewer_tAz1 · 2023-11-01

**Soundness:** 2 fair
**Presentation:** 3 good
**Contribution:** 2 fair
**Rating:** 3
**Confidence:** 5

**Summary:**

The paper presents a detection oriented image-text pre-training to bridge the gap between image-level pre-training and open-vocabulary object detection.
It utilises a simple extension of the contrastive learning method to learn region-level semantic cues.
It also proposes a shifted-window learning approach to improve window attention to make the backbone representation more robust.
Further experiments show good performance on open-vocabulary object detection task on COCO and LVIS.

**Strengths:**

+ The paper is easy to follow.
+ The method demonstrates good performance on rare categories.

**Weaknesses:**

- The proposed contrastive pre-training method is not novel. Incorporate region information in contrastive learning has been explored in many works [1][2][3].
- Using a private data on pre-training is not fair and not convincing.
- The baseline method is already better than previous works. It makes the effectiveness of proposed components questionable.
- Figure 2 is misleading. The similarity map using the backbone features in the middle should be comparing with CLIP based method, not your own.
- In Table 6 (c), stride 4 clearly performs better. The paper doesn't explain why it chooses a suboptimal one.
- Since the method also involves external box annotations, it should compare with previous methods on pure zero-shot OVD benchmark.

[1] RegionCLIP: Region-based Language-Image Pretraining. CVPR 2022

[2] Aligning Pretraining for Detection via Object-Level Contrastive Learning. NIPS 2021

[3] Point-Level Region Contrast for Object Detection Pre-Training. CVPR 2022

**Questions:**

Consider to re-run experiments using public dataset such as CC3M and LAION.

**Details Of Ethics Concerns:**

The paper uses a private dataset to pre-train the model. It is unclear that whether the private data has overlaps with testing benchmarks.

---

### Official Review · Reviewer_Gqbm · 2023-11-01

**Soundness:** 4 excellent
**Presentation:** 3 good
**Contribution:** 4 excellent
**Rating:** 6
**Confidence:** 4

**Summary:**

This submission proposes DITO, a visual-linguistic (VL) pretraining method well
aligned for downstream detection tasks.
CLIP uses global embeddings of images for VL contrastive learning, and thus, it may not be the best choice for localized representation learning and downstream detection.
Here, DITO conducts VL pretraining with a detector-like architecture equipped with
a feature pyramid and detection heads.
During pretraining, the detector heads are trained with ROI features extracted from random regions
due to the lack of bounding box annotations.
Experiments in LVIS OVD show that DITO outperforms previous OVD methods with a margin.

**Strengths:**

- VL pretraining for detection seems a valuable direction of research and
to my knowledge, this work is one of the first attempts along this direction.

- The overall frame is kept simple without using extra annotation or region proposals,
which may inspire the following researches for extensions or applications for new tasks.

- Clear SOTA performance in the large-scale LVIS OVD benchmark.

**Weaknesses:**

- Justification of pretraining with random boxes: I am concerned that the random boxes may produce noisy supervisory signals.
For example, multi-object captions like "a bird and a dog"  may be assigned both for bird regions and dog regions.
I think that some discussions around here are needed.
Did the Authors test other choices here, for example, using regular grids?


- Basic comparisons with CLIP: DITO looks like a VL pretraining method, in essence, rather than an OVD method,
because a significant part of the proposed modifications are in the pretraining phase.
It would be natural to compare it with CLIP in the tasks that it was designed for,
for example, text-sentence matching or zero-shot classification.
Or is it impossible due to architectural changes?

**Questions:**

Please see Weaknesses.

---

### Official Review · Reviewer_rKe8 · 2023-11-04

**Soundness:** 2 fair
**Presentation:** 2 fair
**Contribution:** 2 fair
**Rating:** 5
**Confidence:** 4

**Summary:**

This paper proposes an open vocabulary object detection framework for detection pre-training.

* Unlike previous work using image-level pre-trained VLMs, Dito is based on proposal-level semantic alignment during pre-training.

* Dito consists of three stages. The first stage is image-level alignment based on the Align datasets. The second stage is detection pre-training(or region-semantic pre-training) on this image-text dataset.  The final stage is detection fine-tuning on the base detection dataset.

* The proposed method achieves state-of-the-art performance on the COCO & LVIS dataset benchmark.

**Strengths:**

Overall, the performance of the proposed framework is very good, and the motivation is reasonable.

* I also believe that aligning semantics at the region-level is very important and meaningful. Now many methods are exploring how to achieve region-level pre-training. The core point of open-vocabulary lies in region-language alignment, thereby generalizing to novel classes.

* Due to the framework being based on detection pre-training, it eliminates the need for distillation or pseudo-labeling, which is a strength of this work.

* The comparison experiments are very sufficient, and the proposed method achieves state-of-the-art results on various benchmarks.

**Weaknesses:**

I have several points that I find confusing in this article:

* 1. The entire framework relies on two backbones during inference. The article mentions (there is a tendency to lose the pretrained image-text knowledge, Inspired by previous work (Kim et al., 2023a), we use a separate frozen ViT backbone as an open-vocabulary region classiﬁer at inference time to compute the VLM score z). This indicates that the entire framework does not maintain semantic information during detection pre-training, which seems strange to me and makes the framework appear bulky. If the image-text semantics could be preserved during detection pre-training, this would make the work more reasonable.

* 2. This method uses the aligned dataset, while previous methods follow the clip pretrain setting, making it difficult to compare this method with previous methods. It gives me the feeling that the good results are due to the large amount of data pre-training, rather than Detection-Oriented Pretraining.

* 3. The design of shifted-window learning is strange. It is more of a general network design rather than a solution for open vocabulary detection. Furthermore, shifted-window and open vocabulary seem disconnected.

**Questions:**

* In Table 1, the performance of DITO(ViT-L/16) : mask APr : 40.4, mask AP: 37.7, why the overall mask AP is lower than mask APr, it is really strange here, normally the mask AP(F & C) will be higher than mask APr.

* Overall, the motivation of this paper is reasonable, but it could be more concise. Additionally, the use of a large amount of other data makes it difficult to compare with other works (of course, open vocabulary does require a large amount of image-text data for training). However, I still believe that the experimental setting of this framework is not very clear.